# MOSES: A New Approach to Integrate Interactome Topology and Functional Features for Disease Gene Prediction

**DOI:** 10.3390/genes12111713

**Published:** 2021-10-27

**Authors:** Manuela Petti, Lorenzo Farina, Federico Francone, Stefano Lucidi, Amalia Macali, Laura Palagi, Marianna De Santis

**Affiliations:** Department of Computer, Control and Management Engineering, Sapienza University of Rome, 00185 Rome, Italy; lorenzo.farina@uniroma1.it (L.F.); federicofrancone94@gmail.com (F.F.); lucidi@diag.uniroma1.it (S.L.); macali.1656011@studenti.uniroma1.it (A.M.); laura.palagi@uniroma1.it (L.P.); marianna.desantis@uniroma1.it (M.D.S.)

**Keywords:** disease gene prediction, data integration, precision medicine, computational biology

## Abstract

Disease gene prediction is to date one of the main computational challenges of precision medicine. It is still uncertain if disease genes have unique functional properties that distinguish them from other non-disease genes or, from a network perspective, if they are located randomly in the interactome or show specific patterns in the network topology. In this study, we propose a new method for disease gene prediction based on the use of biological knowledge-bases (gene-disease associations, genes functional annotations, etc.) and interactome network topology. The proposed algorithm called MOSES is based on the definition of two somewhat opposing sets of genes both disease-specific from different perspectives: warm seeds (i.e., disease genes obtained from databases) and cold seeds (genes far from the disease genes on the interactome and not involved in their biological functions). The application of MOSES to a set of 40 diseases showed that the suggested putative disease genes are significantly enriched in their reference disease. Reassuringly, known and predicted disease genes together, tend to form a connected network module on the human interactome, mitigating the scattered distribution of disease genes which is probably due to both the paucity of disease-gene associations and the incompleteness of the interactome.

## 1. Introduction

Precision medicine has been defined as “an emerging approach for disease treatment and prevention that takes into account individual variability in genes, environment, and lifestyle for each person.” [1]. This definition is mainly related to the experimental, methodological, and technological developments of the last decades (e.g., next generation sequencing) that gave birth to new possibilities in the practice of healthcare based on individually tailored therapies. Disease genes identification is an important goal of biomedical research, and one of the main challenges aimed at the development of personalized treatments. In fact, a disease is rarely a consequence of an abnormality on a single gene, but it is usually the result of perturbations involving sets of genes and their relationships (e.g., alteration in molecular interactions, pathways). Disease genes (or seed genes) are those genes whose mutations are involved in diseases, and it is still uncertain whether such genes have unique properties that distinguish them from non-disease genes. In the last decades, numerous databases for gene annotations have been proposed providing information about genes and related diseases. Online Mendelian Inheritance in Man (OMIM) [2], curated by the NCBI and Johns Hopkins University, is one of the most widely used source of information for disease-gene associations, other examples are: PheGenI [3], DisGeNET [4], eDGAR [5]. Despite the several available resources offer different levels of information about the genetic basis of human diseases, knowledge about associations between disease-causing genes and diseases is still incomplete. Moreover, the identification of specific disease genes is often impaired by gene pleiotropy, by the polygenic nature of many diseases, by the influence of a plethora of environmental factors, and by genome variability [6]. Various experimental techniques such as genome-wide association studies (GWAS) and linkage analysis are used to identify new seed genes, but the disadvantage of these high-throughput techniques is that often, they provide long lists of candidate genes and thus require validation procedures that make these methods time-consuming and expensive.

The described open problems combined with the importance of exploiting disease-gene associations to determine personalized treatments paved the way for the development of computational methods. In this context, algorithms for disease gene prediction have been proposed to use and/or integrate the large amount of available omics data and knowledge-based resources (gene annotations, disease-gene associations, etc.). Typical inputs of these algorithms are a set of seed genes (gathered from knowledgebases such as OMIM) and at least a second source of information (protein-protein interactions, functional ontologies, gene expression data, etc.). Instead, the output of these gene prioritization methods are typically subsets of candidate seed genes or genes rankings where top positions are related to high likelihood of involvement in generating a disease phenotype. Several reviews provide a description and a classification of the available algorithms for disease gene prioritization [7,8,9]. Here we briefly describe the three main categories: filtering-based techniques, similarity-based techniques, and network-based techniques. Filtering methods require the definition of filters based on the available knowledge of the molecular basis of the disease under investigation. Similarity-based techniques provide a gene prioritization based on a similarity measure between candidate genes and seed genes: the calculation of the similarity can exploit text-mining approaches [10] and can be based on functional profiles of genes [11]. Finally, network-based methods represent biological data as networks and apply graph mining techniques to rank genes. This last class of algorithms is also the last one developed in time in the wake of the introduction and success of network science in biomedical research [12,13,14,15,16,17]. Several methods have been proposed based on different strategies (network propagation [18,19], module-based [20,21]). Alongside these approaches, recently new network-based methods have been developed that use other genes besides the seed genes to help in the prediction of new disease genes [22,23,24]: in detail, these algorithms exploit genes associated with related diseases.

In this work, we introduce a new method for disease gene prediction based on the use of knowledge-bases, network topological features, and on the k-means algorithm applied to binary data. The proposed algorithm called MOSES (warM and cOld Seeds for disEase geneS) is based on the definition of two different and opposing sets of genes. In fact, for a specific disease, we define the known disease genes as warm seeds, and we identify as cold seeds the genes far from the warm seeds in the interactome (high path length between warm seeds and cold seeds) and not involved in the functions characterizing the disease genes (e.g., molecular pathways). In detail, given the two sets of genes, the key point of the proposed procedure is to distinguish between warm seeds and cold seeds exploiting the topology of the human interactome and the set of functionalities disrupted in the diseases. Regarding the second key point, it is not a priori known how to recognize similarities in functional profile, so that, in practice, we cannot use these similarities to decide if a generic gene is a disease gene or not. To overcome this issue, MOSES is based on the well-known data mining technique *k*-means clustering [25] and exploits an adaptive strategy to guide the clustering procedure to group the majority of known disease genes in a specific cluster. The hypothesis is that this specific cluster contains unknown disease genes (putative disease genes), besides containing known ones. The new approach can exploit and integrate different sources of information and here, we propose a first use based on known disease-gene associations, protein-protein human interactome and two types of functional gene annotations (Gene Ontology terms and *KEGG* pathways).

In the present study, we applied MOSES to a set of 40 diseases. To test the predictive power of MOSES, we performed a computational validation (10-fold cross-validation). Furthermore, we used the enrichment analysis tool Enrichr [26] for checking if the putative genes are enriched in the disease to which the disease genes belong, and we studied the topological features of the predicted disease module (network module composed of known and putative disease genes).

## 2. Methods

The algorithm MOSES (warM and cOld Seeds for disEase geneS) is based on the definition and the characterization of two different and opposing sets of genes: warm seeds (WSs) and cold seeds (CSs). A warm seed is a disease gene, while a cold seed is a gene satisfying two constraints: (i) network-based distance and (ii) functional distance from the warm seeds. The first constraint imposes high path length between WSs and CSs in the interactome (see Figure 1), while the second distance requires that WSs and CSs are involved in totally different biological functions.

Its functioning requires three sequential phases described in detail in the following sections.

### 2.1. Functional Characterization of the Warm Seeds

The first step of the algorithm is to functionally characterize the WSs (i.e., the known disease genes of the disease under investigation) by means of the enrichment analysis (hypergeometric distribution with FDR correction). Different databases can be exploited, and thus integrated, such as Gene Ontology database, *KEGG* pathways, miRTarBase, TRRUST, etc. Fixed a significance threshold, for each of the considered databases, MOSES identifies M significant annotations: only databases for which *M* ≥ 2 are considered for the next steps.

### 2.2. Identification and Enrichment Analysis of the Cold Seeds

To identify the cold seeds, the algorithm first applies the constraint of network-based distance in the selection of a set of genes in the interactome far from the disease genes. Given a specific disease characterized by *P* disease genes (or warm seeds, set *S*_0_) and the interactome *I* composed of *N* genes, the iterative procedure to identify these peripheral genes is described below:
identification of the non-seeds set *NS_i_*. At the first iteration, *NS_1_* is the difference set between the interactome and the disease genes: $NS_{1}=I-S_{0},\;\#\left(NS_{1}\right)=N-P$identification of the first neighbors of genes in *S_i_* (set *FN_i_*)update of the sets *S_i_* and *NS_i_*:$$S_{i}=S_{i-1}\cup FN_{i-1}$$
$$NS_{i}=NS_{i-1}-FN_{i-1}$$


The procedure stops when the ratio between the cardinalities of sets *S*_0_ and *NS_i_* is equal to or greater than 10^−1^. Once the set of peripheral genes has been identified, the MOSES algorithm extracts from this set, the cold seeds selecting only genes not involved in the WSs significant annotations (GO terms, *KEGG* pathways, MicroRNA-Target interactions, transcription factor-target regulatory relationships, etc.).

To be consistent with the WSs characterization, the algorithm selects also for the CSs the first M annotations of the considered databases with smaller *p*-values according to the hypergeometric distribution with FDR correction. Now, for each type of annotation, a set denoted by *J*, made of 2*M* annotations, can be built: the first *M* terms functionally describe the WSs, while the second half is related to the CSs characterization.

### 2.3. Optimized Clustering Phase and Selection of Putative Disease Genes

The set *J* is the input of the clustering phase: it is used to identify the subset of *G_J_* genes, namely the genes in the interactome involved in the selected annotations, and allows to build the *G_J_*-by-2*M* matrix to be subjected to the clustering procedure. This step is based on the use of the popular *k*-means clustering algorithm [25]: let *k* be a fixed integer number, the *k*-means separates the input set of genes into *k* clusters. MOSES algorithm proceeds as follows. Starting from *k* = 2, it iteratively applies the *k*-means clustering algorithm to the *G_J_*-by-2*M* matrix, until it identifies a reasonable value for *k*. Note that, in the first iteration of the algorithm, namely when *k* = 2, it is likely that the known disease genes belonging to *G_J_* are grouped within the same cluster, due to the specific choice of the set *J*. Then, the process goes on increasing the number *k* of clusters incrementally by one. The algorithm ends up at the first iteration for which the number of clusters *k_max_* is the maximum number of clusters so that a given percentage *q*% of disease genes within *G_J_* is in the same cluster *C^*^*. This percentage belongs to the range (60%, 90%): the threshold of 60% is set to obtain more than half of the disease genes in one out of *k* clusters. In the case, at the first iteration (*k* = 2), the disease genes within *G_J_* are divided into two clusters containing each less than 90% of them, MOSES sets *q*% equal to the higher percentage only if *q*% ≥ 60%. Note that WSs e CSs do not share any annotations by construction, hence the optimal cluster cannot contain both the type of seeds.

The procedure outlined above is repeated considering each database (Gene Ontology database, *KEGG* pathways, *miRTarBase*) and allows to obtain the sets of genes *C_i_^*^* (*i* = GO; *KEGG*; miRTarBase). The new algorithm performs the intersection among the sets *C_i_^*^* returning a batch of known and putative disease genes.

The above-described procedure is synthesized in Figure 2.

## 3. Data and Preprocessing

In the present work, to avoid selection bias, we applied the MOSES algorithm to 40 diseases selected from those provided in [21]. The selection criterion is related to the number of disease genes (warm seeds set, *S*_0_): #(*S*_0_) = P, P∈(25,150). As described in detail in [21], the disease-gene associations were retrieved from OMIM (Online Mendelian Inheritance in Man; http://www.ncbi.nlm.nih.gov/omim [2], accessed on 26 April 2019) and from the PheGenI database (Phenotype-Genotype Integrator; http://www.ncbi.nlm.nih.gov/gap/PheGenI [3], accessed on 26 April 2019). We used the human protein–protein interactome provided in [27] (243,603 protein-protein interactions connecting 16,677 unique proteins) and we considered two kinds of annotations: GO terms (Gene Ontology database, biological process, downloaded 26 April 2019) and pathways (*KEGG* gene set from the Molecular Signatures Database, version 6.2). The available GO terms (biological process) were not propagated upwards on the GO tree and were prefiltered as follows [20]:
annotations labeled with evidence code IPI (Inferred from Physical Interaction) were excluded to avoid circularity;annotations not associated with the gene products (evidence code “NOT”) were excluded.


## 4. Results and Discussion

MOSES algorithm is based on the definition of two different and opposing sets of genes (warm seeds and cold seeds) and its functioning required the above described sequential phases. The putative disease genes returned by the algorithm are characterized by two important properties: the network-based proximity and the functional similarity with the original disease genes (here defined warm seeds). This is possible thanks to the new definition of the cold seeds: genes far from the disease genes in the interactome and not involved in their functions. Furthermore, it is worth noting that MOSES can exploit, and thus integrate, different sources of information.

As described in the previous section, we applied MOSES to 40 diseases and for the first step of the algorithm (functional characterization of the WSs by means of the enrichment analysis), we set the significant threshold equal to 5 × 10^−2^: for 27 out of the 40 diseases, MOSES selected *M* ≥ 2 significant annotations in both cases (GO-BP terms, *KEGG* pathways). As MOSES has been thought to exploit data integration in the prediction of new disease genes, we considered only the subset of 27 diseases and in Table 1, we show for all of them: the number of WSs, the number of genes identified by MOSES applying the first constraint of network-based distance (peripheral genes) and the number of CSs (i.e., peripheral genes functionally distant from the WSs). It is worth noting that the application of the functional distance constraint further filters the set of peripheral genes proving that the integration between protein-protein interactome topology and gene functional annotations databases allows to appropriately identify the two opposing sets of gens.

For the optimized clustering phase, we used the *k*-means algorithm implementation in Matlab (*k*-means++ algorithm), setting as input parameters hamming distance and 50 replicates (number of times the *k*-means algorithm is run with different centroids).

In Figure 3 we show the clustering phase application to amino acid metabolism inborn errors, characterized by 52 WSs and 113 CSs. For the warm seeds, MOSES selected *M* = 25 significant GO-BP terms (*p*-value < 5 × 10^−2^ according to the hypergeometric test with FDR correction) leading to the set *J* composed of 2*M* = 50 annotations (the second half of them is related to the cold seeds functional characterization). 1594 genes (32 of which are disease genes) of the interactome are involved in the 50 selected annotations. At the first iteration, the *k*-means algorithm produces two clusters with 1345 and 249 genes, respectively. One of the clusters contains 100% of disease genes, thus the process goes on increasing the number of clusters *k* and stops with *k_max_* = 3, as one cluster contains the largest percentage of seeds equal to 0.84% (27 out of 32 disease genes, see Figure 3). 

Considering *KEGG* database, for the warm seeds, the algorithm selected *M* = 10 significant pathways (*p*-value < 5 × 10^−2^ according to the hypergeometric test with FDR correction) leading to the set *J* composed of 2*M* = 20 terms. The genes involved in the 20 selected pathways are 978, 29 of them are disease genes. In this case, the process goes on increasing the number of clusters *k* until the selection of *k_max_* = 5, as it is the first iteration for which we obtain a cluster containing the 86.2% of the disease genes (see Figure 3).

Thus, for this disease, we obtain the cluster $C_{GO}^{*}$ made of 1184 genes (27 out of them being disease genes) and the cluster $C_{KEGG}^{*}$ made of 387 genes (25 out of them being disease genes) The intersection $C_{GO}^{*}\cap C_{KEGG}^{*}$, returns 138 genes: 16 disease genes and 112 putative disease genes (set PG).

In 5 diseases among the 27 studied, the clustering phase failed in the identification of the cluster *C^*^* using at least one type of annotations (GO-BP terms, *K_EGG_* pathways). For example, in the case of asthma, at the first iteration (*k* = 2) the k-means algorithm applied to GO-BP data, returns 2 clusters containing 57% and 43% of disease genes: the warm seeds are therefore divided into two halves. In these cases, the use of a third (or more) database(s) could help to overcome the limitation encountered with a specific type of gene annotations. For the other 22 diseases, the MOSES algorithm identified the set of putative disease genes (*PG*).

### 4.1. Computational Cross-Validation and Comparison with Random Walk with Restart

To test the predictive power of MOSES, we performed the 10-fold cross-validation. For each disease, we randomly split the disease genes set *S*_0_ into 10 subsets. Each time, we selected one subset as probe set *S_P_* and the rest nine subsets as training warm seeds set *S_T_*. Then we measured MOSES ability to recover genes in *S_P_*. Furthermore, to evaluate the relative performance of MOSES, we considered as a reference another algorithm for candidate gene prioritization. We selected the random walk with restart algorithm (RWR) [18]: it is a ranking algorithm exploiting global network topology and it was shown to outperform other methods [19].

For each disease, we applied RWR (restart probability *r* = 0.7) to the same training sets used with MOSES, and to make comparable the outputs of the two algorithms (a finite set of *PG* putative disease genes for MOSES and a genes ranking for RWR), we considered the top *PG* positions in the case of RWR. As a measure of performance, we considered the percentage of recovered warm seeds in the test set *SP*. In Figure 4, we show in detail the difference between the two algorithms: we can appreciate the tendency of MOSES to outperform RWR in most cases, however, in few cases, RWR works better than MOSES. As expected indeed, there is not a universal best algorithm, but in general, the selection of the algorithm should be taken considering different factors of the available data. However, results of the statistical comparison (paired *t*-test) between the two algorithms show that overall MOSES performances are significantly higher than overall RWR performances (*p*-value = 9.6 × 10^−03^).

### 4.2. Enrichment Analysis of Putative Disease Genes

We used the enrichment analysis tool Enrichr [23] to check if the putative genes are enriched in the disease to which the disease genes belong (category: Diseases/Drugs, section: *DisGeNET*). To find the corresponding disease in Enrichr, we referred to the International Statistical Classification of Diseases (ICD-11 for Mortality and Morbidity Statistics, version: 09/2020). Results are shown in Table 2. For 19 out of 22 diseases, the adjusted *p*-value is below the threshold of 0.05. Only in one case (*hemolytic anemia*), the *p*-value is above the significance threshold, while for two diseases (carbohydrate metabolism inborn errors, lipid metabolism disorders) we did not find the corresponding disease in Enrichr.

### 4.3. Study of the Predicted Disease Module

Putative disease genes are identified by MOSES exploiting the protein-protein interactome topology and the set of functionalities disrupted in the diseases. The integration of these different types of information agrees with the hypothesis of overlap among disease module, topological module (locally dense network neighborhood) and functional module (aggregation of nodes with similar or related functions in the same network neighborhood) [13]. While the topological and functional modules are concepts widely applied in different fields and suitable also in the case of biological networks, the network disease module is a recent key concept of network medicine [13,28,29]. This concept was raised from some broadly accepted hypotheses and organizational principles of disease genes [30]. In particular, genes (or gene products) involved in the same disease tend to interact (local hypothesis) and to cluster in connected subnetworks (*disease module hypothesis*). Moreover, genes in a disease module are often involved in the same biological functions (*functional coherence hypothesis*).

In the light of these considerations, for each disease, we studied the topology of the network module composed of the known disease genes (the warm seeds, set *S*_0_) and the candidate genes (set *PG*) suggested by MOSES. We focused on the largest connected component (*LCC*) of the disease module investigating the size of the *LCC* consisting of warm seeds only ($\left|LCC_{WS}\right|$), the size of the *LCC* considering the set $S_{0}\cup PG$ ($\left|LCC_{WS+PG}\right|$) and the number of warm seeds in $LCC_{WS+PG}$. In Table 3, we show the above describe measures for each disease.

In all the cases except for lysosomal storage diseases, the largest connected component of the predicted disease module (known and putative disease genes) contains a higher number of WSs with respect to the size of the *LCC* composed of warm seeds only. This result suggests that the extension of the disease module with the identified candidate genes, mitigates the WSs scattered distribution in the human interactome. The obtained disease modules are thus in accordance with the strategy of the recent Seed Connector Algorithm (SCA) [31]. Indeed, SCA proposes to add few additional linking genes (seed connectors) to the disease genes set, on the basis of the hypothesis that such seed connectors are hidden disease module elements that are critical for interpreting the functional context of disease genes [31]. However, the main and substantial difference between the two algorithms is that SCA builds the network module forcing the presence of the disease genes connected component, while with the application of MOSES, this topological property of the disease module is the result of the algorithm procedure. Furthermore, MOSES does not impose the presence of a single connected component.

Figure 5 shows the case of *ulcerative colitis* (see Appendix A for the other diseases). For this disease, the largest connected component of *warm seeds* only is composed of 5 nodes: thus only 5 out of 56 known disease genes are directly connected in the human interactome. Adding the 165 putative genes suggested by the MOSES algorithm to the disease module, we obtain the largest connected component composed of 140 nodes, 22 of which are WSs.

To verify that the obtained size of the $LCC_{WS+PG}$ has not been obtained by chance, for each disease, we generated 1000 times a random disease module composed of the known disease genes and genes randomly selected from the interactome (*WSs* were excluded during random picks). In particular, the number of random genes (*RG*) is equal to the number of putative disease genes. For all the diseases except for *lysosomal storage diseases*, the size of the $LCC_{WS+PG}$ is above the threshold of 95^th^ percentile of the distribution of the largest connected components composed of the known disease genes and the random genes ($\left|LCC_{WS+RG}\right|$, see Table 3).

### 4.4. Case Studies on Colorectal Neoplasms and Rheumatoid Arthritis

Among the studied diseases, we present in this section the results obtained for rheumatoid arthritis and colorectal neoplasms.

#### 4.4.1. Rheumatoid Arthritis

Rheumatoid arthritis (RA) is an autoimmune and inflammatory disease, which means that the immune system attacks healthy cells by mistake, causing inflammation (painful swelling) in the affected parts of the body.

The disease genes for RA used in this work (Appendix A), are enriched in 18 KEGG pathways (hypergeometric test, FDR less than 0.05): among them, it is worth noting the presence of notch signaling pathway [32,33], cell adhesion molecules CAMs [34] and Jak-STAT pathway [35].

Starting from the 42 original disease genes, the MOSES algorithm identified 447 putative genes (Appendix A). The functional enrichment analysis showed they are enriched in 41 KEGG pathways (hypergeometric test, FDR less than 0.05), including 16 of the 18 characterizing the disease genes. However, none of them are in the top 3 (FDR ascending order). Indeed, the putative genes resulted mainly associated with neuroactive ligand receptor interaction (FDR = 1.44 × 10^−43^), olfactory transduction (FDR = 7.82 × 10^−20^) and metabolism of xenobiotics by cytochrome P450 (FDR = 4.28 × 10^−19^). Interestingly, in relation to the olfactory transduction pathway, disturbances in the olfactory function have been investigated mainly in neurological/neurodegenerative disorders and only recently in autoimmune diseases [36,37,38]. In particular, in [39], Li and colleagues carried out a whole-exome sequencing study in a Han (Chinese ethnic group) patient cohort and identified genes enriched in the olfactory transduction pathway, suggesting the potential involvement of this pathway in RA disease progression.

Furthermore, performing the enrichment analysis with Enrichr (category: transcription, section: TRRUST Transcription Factors 2019) and focusing on the top 3 positions (ascending order based on adjusted *p*-value), we found that the putative genes are significantly enriched in RelA (adjusted *p*-value = 1.104 × 10^−14^), NF-κB1 (adjusted *p*-value = 4.502 × 10^−14^) and CIITA (adjusted *p*-value = 4.734 × 10^−12^). NF-κB is a collective name for dimeric transcription factors comprised of the Rel family of proteins that include RelA (p65), c-Rel, RelB, NF-κB1 (p50), and NF-κB2 (p52): NF-κB has been well recognized as a pivotal regulator of inflammation in rheumatoid arthritis [40].

#### 4.4.2. Colorectal Neoplasms

Colorectal cancer is one of the most common cancers in the world and also one of the leading causes of cancer-related death worldwide [41].

The disease genes here used (Appendix A), are enriched in 19 K_EGG_ pathways (hypergeometric test, FDR less than 0.05): among them, as expected, there are colorectal cancer pathway, pathways in cancer, and mismatch repair. For this disease, starting from the 42 original disease genes, MOSES suggested 1160 putative genes (Appendix A). The functional enrichment analysis showed they are enriched in 65 KEGG pathways (hypergeometric test, FDR less than 0.05), including almost all those characterizing the disease genes (16 out of 19 terms). It is worth noting that only for the putative genes and at the first position in their pathways ranking based on FDR ascending order, we found cytokine-cytokine receptor interaction (FDR = 4.18 × 10^−63^). Indeed, cytokine and cytokine receptor interaction networks are crucial aspects of inflammation and tumor immunology particularly for colorectal cancer [42,43]. Moreover, using the Enrichr platform (category: transcription, section: miRTarBase 2017), we found that putative disease genes are enriched in mir-145-5p (adjusted *p*-value = 4.97 × 10^−9^; top position in the ascending order). miR-145 has frequently been investigated in colorectal cancer [44]: this miRNA acts as a tumor suppressor [45,46] and has been reported to be down-regulated in colon carcinomas [47].

## 5. Conclusions

In this work, we introduce the algorithm MOSES based on the new definition of warm seeds and cold seeds. In particular, the identification of the cold seeds requires the application of two constraints of distance from the known disease genes (here defined warm seeds): network-based distance and functional distance. MOSES exploits thus the advantages of network-based approaches and the use of disease genes functional features: indeed, it suggests a finite set of putative disease genes characterized by the two important properties of network-based proximity and functional similarity with the original disease genes. The use of these two seeds sets is innovative in the fact that we consider genes far away from each other to identify putative genes, whereas most disease gene prediction algorithms are based on the idea that putative genes are “near” in some sense to the known disease genes. Future analysis will be aimed at the integration of more types of gene annotations, to overcome the limitation encountered in the present study.

## Figures and Tables

**Figure 1 genes-12-01713-f001:**
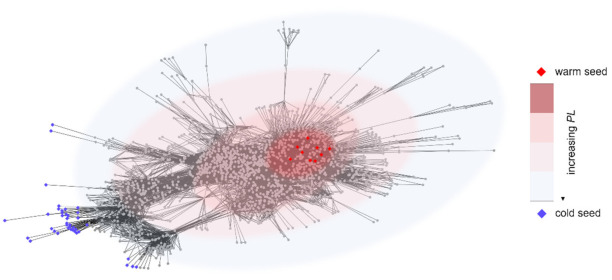
Example of network-based distance between warm seeds (red diamonds) and cold seeds (blue diamonds). The color of background ovals codes for the path length (PL) between WSs and CSs.

**Figure 2 genes-12-01713-f002:**
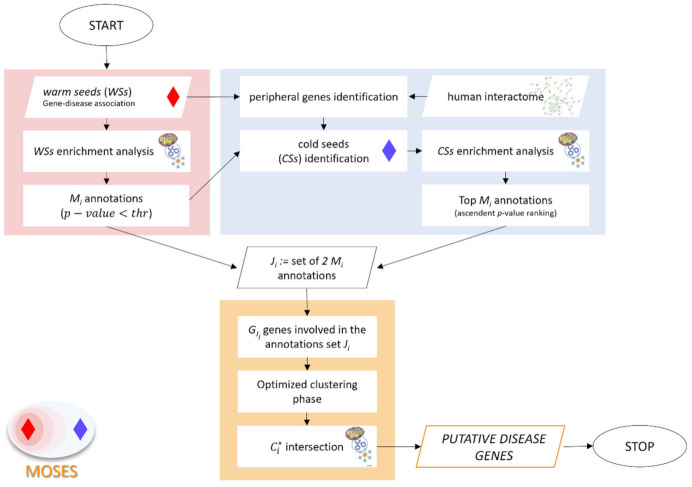
Flowchart of MOSES algorithm. The background rectangles identify the three sequential phases: red, blue and orange respectively for: (1) WSs functional characterization, (2) CSs identification and characterization, and (3) optimized clustering phase. The enrichment analysis can be performed considering different types of gene annotations (Gene Ontology database, KEGG pathways, miRTarBase) and obtaining for each of them *M_i_* annotations, with *i* = GO; KEGG; miRTarBase, etc.

**Figure 3 genes-12-01713-f003:**
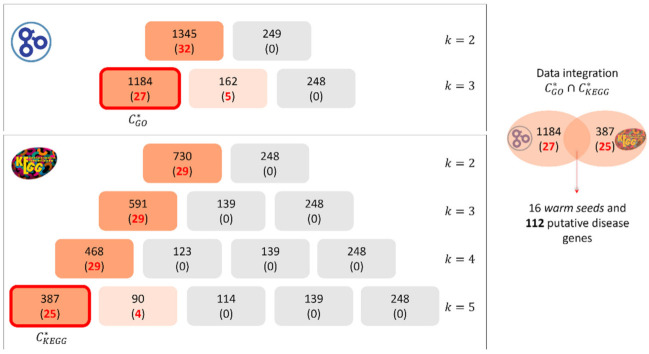
Clustering process applied to the disease amino acid metabolism inborn errors using GO-BP annotations (top panel) and KEGG pathways (bottom panel). On the right, the procedure of data integration and the identification of putative disease genes are shown.

**Figure 4 genes-12-01713-f004:**
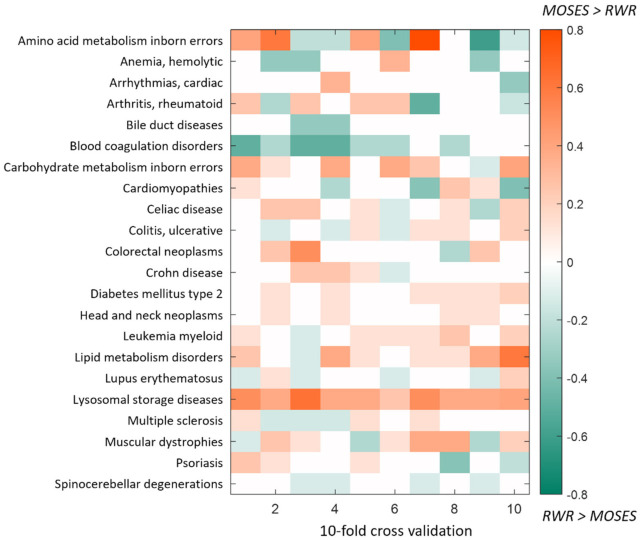
10-fold cross-validation. Difference between MOSES and RWR performances. The performances are computed as the percentage of recovered warm seeds in the test set *S_P_*. Rows and columns represent respectively the diseases and the cross-validation iterations. In the case of positive values (orange pixels), MOSES outperforms RWR, while negative values (green pixels) refer to the opposite situation.

**Figure 5 genes-12-01713-f005:**
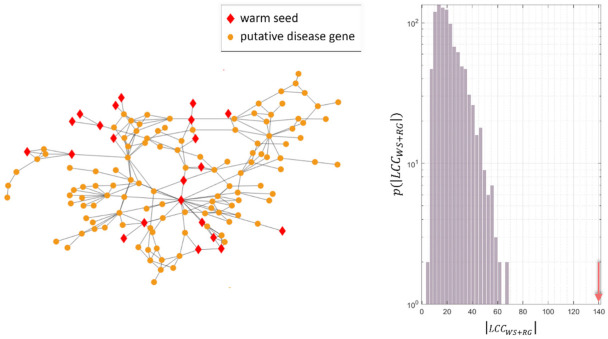
Largest connected component (*LCC*) of the predicted disease module (*warm seeds* and putative disease genes) for *ulcerative colitis*. Node shape codes for the type of genes: red diamonds represent the *warm seeds* (22 nodes), while orange dots represent the putative disease genes (118 nodes). On the right, distribution of the size of the 1000 *LCCs* of the random disease modules ($\left|LCC_{WS+RG}\right|$) obtained adding to the *warm seeds*, a set of randomly selected genes with cardinality equal to the set of putative genes; the orange arrow indicates the size of the $LCC_{WS+PG}$ shown in the left panel.

**Table 1 genes-12-01713-t001:** Cardinalities of the 3 genes sets: WSs, peripheral genes (application of network-based distance constraint) and CSs (application of both network-based and functional distance constraints).

Disease	Warm Seeds	Peripheral Genes	Cold Seeds
Amino acid metabolism inborn errors	52	119	113
Anemia, hemolytic	29	155	143
Arrhythmias, cardiac	30	171	163
Arthritis, rheumatoid	42	87	77
Asthma	37	91	85
Bile duct diseases	31	109	103
Blood coagulation disorders	40	142	129
Blood platelet disorders	26	193	170
Carbohydrate metabolism inborn errors	77	81	79
Cardiomyopathies	50	70	63
Celiac disease	36	137	120
Colitis, ulcerative	56	90	72
Colorectal neoplasms	42	79	68
Crohn disease	72	65	56
Diabetes mellitus, type 2	73	77	75
Head and neck neoplasms	35	87	80
Leukemia, myeloid	43	97	93
Lipid metabolism disorders	50	93	83
Lung diseases, obstructive	40	88	82
Lupus erythematosus	75	51	48
Lysosomal storage diseases	45	152	150
Multiple sclerosis	69	71	62
Muscular dystrophies	36	113	107
Psoriasis	54	86	76
Renal tubular transport inborn errors	34	229	211
Spinocerebellar ataxias	28	147	132
Spinocerebellar degenerations	30	147	137

**Table 2 genes-12-01713-t002:** Enrichment analysis of putative disease genes performed with Enrichr (Diseases/Drugs category, DisGeNET section). For each disease, we show the number of putative disease genes (PG), the corresponding DisGeNET disease name, the number of validated PG and the adjusted *p*-value retrieved from Enrichr; *p*-values below the significance threshold are highlighted in red.

Disease	#PG	DisGeNET Disease	#Validated	Adjusted *p*-Value
Amino acid metabolism, inborn errors	122	Amino Acid Metabolism, Inborn Errors	2	1.68 × 10^−02^
Anemia, hemolytic	50	Anemia, Hemolytic	2	7.52 × 10^−02^
Arrhythmias, cardiac	59	Cardiac Arrhythmia	5	7.08 × 10^−04^
Arthritis, rheumatoid	447	Rheumatoid Arthritis	156	7.92 × 10^−49^
Bile duct diseases	55	Bile Duct Diseases	1	3.71 × 10^−02^
Blood coagulation disorders	104	Blood Coagulation Disorders	13	9.73 × 10^−10^
Carbohydrate metabolism inborn errors	256	-	-	-
Cardiomyopathies	32	Cardiomyopathies	22	1.04 × 10^−04^
Celiac disease	112	Celiac Disease	16	4.16 × 10^−10^
Colitis, ulcerative	165	Ulcerative Colitis	68	7.88 × 10^−45^
Colorectal neoplasms	1160	Colorectal Carcinoma	433	2.13 × 10^−84^
Crohn disease	162	Crohn Disease	58	3.50 × 10^−34^
Diabetes mellitus, type 2	52	Diabetes Mellitus, Non-Insulin-Dependent	29	4.68 × 10^−15^
Head and neck neoplasms	412	Malignant Head and Neck Neoplasm	52	1.21 × 10^−21^
Leukemia, myeloid	184	Myeloid Leukemia	22	3.32 × 10^−08^
Lipid metabolism disorders	43	-	-	-
Lupus erythematosus	248	Lupus Erythematosus, Systemic	103	1.19 × 10^−59^
Lysosomal storage diseases	112	Lysosomal Storage Diseases	5	3.18 × 10^−03^
Multiple sclerosis	396	Multiple Sclerosis	101	1.31 × 10^−37^
Muscular dystrophies	122	Muscular Dystrophy, Duchenne	6	6.63 × 10^−03^
Psoriasis	421	Psoriasis	77	3.51 × 10^−27^
Spinocerebellar degenerations	38	Ataxia, Spinocerebellar	2	3.52 × 10^−02^

**Table 3 genes-12-01713-t003:** Study of the largest connected component (LCC) of the disease module. For each disease, we show the number of warm seeds (WSs), the number of putative disease genes (PG), the size of the LCC consisting of warm seeds only ($\left|LCC_{WS}\right|$), the size of the LCC considering the set $S_{0}\cup PG$ ($\left|LCC_{WS+PG}\right|$) , the number of warm seeds in $LCC_{WS+PG}$, the 95th percentile threshold of the distribution of the 1000 LCCs of the random disease module (known disease genes and random genes). In the column $\left|LCC_{WS+PG}\right|$, bold text highlights values above $\left|LCC_{WS+RG}\right|$ threshold.

Disease	#WSs	#PGs	$$\left|LCC_{WS}\right|$$	$$\left|LCC_{WS+PG}\right|$$	$$\#\mathrm{WSs}\;\mathrm{in}\;LCC_{WS+PG}$$	$$\left|LCC_{WS+RG}\right|\;\mathrm{Threshold}$$
Amino acid metabolism inborn errors	52	122	11	**42**	14	27
Anemia, hemolytic	29	50	11	**55**	12	16
Arrhythmias, cardiac	30	59	2	**36**	6	16
Arthritis, rheumatoid	42	447	6	**306**	31	201
Bile duct diseases	31	55	3	**35**	7	12
Blood coagulation disorders	40	104	22	**98**	34	37
Carbohydrate metabolism inborn errors	77	256	9	**168**	39	96
Cardiomyopathies	50	32	27	**42**	32	33
Celiac disease	36	112	2	**57**	7	15
Colitis, ulcerative	56	165	5	**140**	22	44
Colorectal neoplasms	42	1160	18	**992**	35	771
crohn disease	72	162	10	**150**	27	57
Diabetes mellitus type 2	73	52	7	**19**	9	16
Head and neck neoplasms	35	412	6	**320**	25	172
Leukemia myeloid	43	184	16	**136**	32	69
Lipid metabolism disorders	50	43	11	**37**	19	17
Lupus erythematosus	75	248	5	**180**	39	92
Lysosomal storage diseases	45	112	8	13	5	20
Multiple sclerosis	69	396	11	**287**	40	185
Muscular dystrophies	36	122	12	**84**	24	31
Psoriasis	54	421	6	**309**	36	194
Spinocerebellar degenerations	30	38	2	**37**	9	12

## Data Availability

The data presented in this study as well as the code used are available at https://github.com/ManuelaPetti/MOSES.git (accessed on 24 May 2021).

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
