# Peer review of "MOSES: A New Approach to Integrate Interactome Topology and Functional Features for Disease Gene Prediction"

_genes, 2021, doi:10.3390/genes12111713_

Round 1

Reviewer 1 Report

The authors developed a network-based algorithm for diseases gene prediction by considering two perspectives, the cold and the warm genes for disease gene prediction.   In section 3.1, where RWR also outperforms MOSES, can the authors give a possible explanation why this could be the case for the datasets this occurs for especially because MOSES considers two opposing sets of genes for its prediction?     As a follow-up from above,  is using this two gene perspective for prediction only significant for some diseases? Which ones and what are the general characteristics? Basically, is there a pattern?   Can the authors make the source code for the MOSES algorithm open source?   Also, the data preprocessing method is not described. This should be added to the improved manuscript

Reviewer 2 Report

The authors present an original method to predict disease associated genes. They test their method with 40 diseases, evaluating the recall by cross-validation and comparing with a state of the art method (random walk with restart). The paper is clearly presented. I agree that this work deserves to be published after the clarification of some aspects.

Major comments:

1- One of the novelties of the proposed method is the use of additional genes (the cold seeds) besides the known disease genes (warm seeds), to improve the predictions. The authors should acknowledge in the introduction that other methods have been developed that seek other genes besides de warm seeds to help in the prediction of new disease genes, such as: 

-Garcia-Vaquero, M. L., Gama-Carvalho, M., Rivas, J. D. L. & Pinto, F. R. Searching the overlap between network modules with specific betweeness (S2B) and its application to cross-disease analysis. Scientific Reports 8, 11555 (2018).  

-Cáceres, J. J. & Paccanaro, A. Disease gene prediction for molecularly uncharacterized diseases. PLoS Computational Biology 15, e1007078-14 (2019).  

-Maiorino, E. et al. Discovering the genes mediating the interactions between chronic respiratory diseases in the human interactome. Nat Commun 11, 811 (2020).

These methods are different from the presented method, as they look for information in genes associated with related diseases, while the proposed method tries to identify a set of genes that is as unrelated with the warm seeds as possible. 

2- The authors should present absolute values of performance metrics and not only the difference to a competing approach (as they do in figure 4). That is, they should present recall and precision values for their method and for RWR, even if only in supplementary material. This is important to fully evaluate your method and to compare with other (past and future) competing approaches. 

3- I do not fully understand why the final cluster cannot contain 100% of the warm seeds. You could choose the number of clusters in order to maximize the percentage of included warm seeds and minimize the total number of genes in the cluster while including genes other than just warm seeds.   

4- The authors choose to intersect the results obtained with the GO and the KEGG annotations. The idea is that the final prediction is going to be more robust. However, GO and KEGG annotations cover domains that are not completely overlapping (some cellular functions are better covered at KEGG while others are only considered in GO). Therefore I am not sure that the best option is the intersection. The authors should compare the performance of the intersection with the union of GO and KEGG results or with a single clustering applied on a combined matrix that joins GO and KEGG annotations. As usual, the best option can change with the disease, but the authors can suggest the approach with the best average, or the one with less variable cross-validation results. 

Reviewer 3 Report

In their manuscript "MOSES: A New Approach to Integrate Interactome Topology 2 and Functional Features For Disease Gene Prediction” Petti et al. explore disease gene prediction through analysis of protein-protein interaction networks. They try to conceptualize a framework where known disease genes representing “hot seeds” and genes distant in the network (“cold seeds”) are used to identify clusters of new disease associations.

The manuscript is not written well, instead it is extremely repetitive and has a very unusual style which resembles typical writing in online chats (e.g. frequent usage of “…” in lists). This makes the manuscript barely readable.

In addition, the authors do not provide any implementation of their “algorithm”. There is no webtool implementation or code repository to allow the scientific community to use and test the approach.

Finally, the whole concept is circular. The authors use disease annotation and interaction databases, apply clustering to these and then validate their results with disease annotation databases again. The authors make no attempt to enable reproducibility of their results. The whole supplementary data is basically two gene name lists and a repletion of the same figure of networks with basically the same legend. For the p-value calculations they do not control for any obvious confounders like gene size, interaction count or amount of published literature to name a few.

Round 2

Reviewer 1 Report

None

Reviewer 2 Report

The authors have addressed all my comments.

Reviewer 3 Report

The authors made no attempt to address my comments "e.g. the writing is still below average and the ... are still included).

The GitHub link provided looks like MATLAB scripts drag and droped to a newly created repository. It is basically 3 scripts with merely ~600 lines of procedural code (two scripts for the matrix were later added). It even has the work directories hard coded ('M:\Network Medicine\DA&R\Results_DiaBLe'), which poses a security risk for the institute. The data is missing (e.g. PPI201806_large.txt). A Readme and installation instruction is missing. If a specific MATLAB version and plugins are required this should be stated somewhere and the authors should provide a running version of their tool (maybe in a Docker container).

The code and repository thus do not allow reproducibility and do not enable usage by anybody.